# Quantitative Detection of Mastitis Factor IL-6 in Dairy Cow Using the SERS Improved Immunofiltration Assay

**DOI:** 10.3390/nano12071091

**Published:** 2022-03-26

**Authors:** Ruipeng Chen, Hui Wang, Yiguang Zhao, Xuemei Nan, Wensong Wei, Chunmei Du, Fan Zhang, Qingyao Luo, Liang Yang, Benhai Xiong

**Affiliations:** 1State Key Laboratory of Animal Nutrition, Institute of Animal Science, Chinese Academy of Agricultural Sciences, Beijing 100193, China; wanghui_lunwen@163.com (H.W.); zhaoyiguang@caas.cn (Y.Z.); xuemeinan@126.com (X.N.); duchunmeim@163.com (C.D.); zhangfan19@139.com (F.Z.); luoqingyao@caas.cn (Q.L.); 2Institude of Food Science and Technology, Chinese Academy of Agricultural Sciences, Beijing 100193, China; weiwensong@caas.cn

**Keywords:** cow mastitis, interleukin-6 (IL-6), immunofiltration assay (IFA), SERS nanotag

## Abstract

Interleukin-6 (IL-6) is generally used as a biomarker for the evaluation of inflammatory infection in humans and animals. However, there is no approach for the on-site and rapid detection of IL-6 for the monitoring of mastitis in dairy farm scenarios. A rapid and highly sensitive surface enhanced Raman scattering (SERS) immunofiltration assay (IFA) for IL-6 detection was developed in the present study. In this assay, a high sensitivity gold core silver shell SERS nanotag with Raman molecule 4-mercaptobenzoic acid (4-MBA) embedded into the gap was fabricated for labelling. Through the immuno-specific combination of the antigen and antibody, antibody conjugated SERS nanotags were captured on the test zone, which facilitated the SERS measurement. The quantitation of IL-6 was performed by the readout Raman signal in the test region. The results showed that the detection limit (LOD) of IL-6 in milk was 0.35 pg mL^−1^, which was far below the threshold value of 254.32 pg mL^−1^. The recovery of the spiking experiment was 87.0–102.7%, with coefficients of variation below 9.0% demonstrating high assay accuracy and precision. We believe the immunosensor developed in the current study could be a promising tool for the rapid assessment of mastitis by detecting milk IL-6 in dairy cows. Moreover, this versatile immunosensor could also be applied for the detection of a wide range of analytes in dairy cow healthy monitoring.

## 1. Introduction

Mastitis is a widespread disease in dairy cows, which not only great restricts the development of the dairy industry, but also causes heavy economic losses in the dairy farming industry. Due to the complex pathogenesis and low cure rate, the incidence of dairy cow mastitis could be as high as 50~70% in China, and the treatment cost is nearly 200 million yuan every year [1]. Many researchers are devoted to investigating the underlying biological pathogenesis of mastitis to diagnose and manage this disease [2,3,4]. Pathogen infection is the main cause of mastitis. When this occurs, cytokines are released to participate in the immune responses and regulate the local inflammatory reactions [5]. Moreover, due to a rapid increase in milk yield in the early lactation stage of dairy cows, mammary epithelial cells usually secrete some proinflammatory cytokines and trigger immune responses [6]. Consequently, cow mastitis is closely related to the immune response promoted by inflammatory cytokines.

Cytokines play important roles in the immune system of dairy cows by regulating various cell physiological functions. In addition, they act as transmitters of cell communication in the process of inflammation and immune response [7,8]. As a multifunctional cytokine, interleukin-6 (IL-6) is a kind of glycoproteins produced and secreted by activated macrophages, lymphocytes, and epithelial cells. When dairy cows have bacterial infections such as mastitis and endometritis, a lot of inflammatory and immunologic cells in the body can express the IL-6 gene and produce IL-6 [9,10]. Therefore, IL-6 is often used clinically as a monitoring index and as an evaluation standard for mastitis in dairy cows. Under normal circumstances, the IL-6 concentration in healthy dairy cows is less than 13.02 and 164.47 pg mL^−1^ in the serum and milk, respectively. When mastitis occurs, the IL-6 content in the serum and milk of dairy cows will increase significantly up to 254.32 and 432.09 pg mL^−1^, respectively [11,12]. So, the concentration of IL-6 in the milk of dairy cows can practically reflect the incidence of the inflammatory response [13,14]. Therefore, the rapid and high-sensitivity detection of IL-6 is of great significance in the early prevention and diagnosis of inflammatory diseases such as mastitis in dairy cows. On the contrary, delaying diagnosis and treatment will greatly increase the treatment cycle and medical cost [15].

At present, the commonly used methods for detecting the inflammatory biomarker IL-6 in dairy cows mainly include enzyme-linked immunoassay (ELISA) [16], chemiluminescence immunoassay (CLIA) [17], leukocyte count [18], and electrochemical immunosensor [19]. However, some limitations still exist in these methods, such as expensive equipment, time-consuming, complex procedures, and narrow linear range of results. These results are insufficient for the effective diagnosis of inflammatory diseases in dairy cows. It is urgent and challenging to establish a simple and easy method to quantitatively detect IL-6 in milk.

As an on-site real-time detection method, the immunofiltration assay (IFA, immunofiltration kit) has the advantage of having a simple structure, convenient operation, quick analysis, and low cost. It has been widely used in the field of inflammatory biomarker detection [20]. Nevertheless, the traditional colloidal gold-based assay uses color change to discriminate target concentration, which greatly limits the detection accuracy and sensitivity [21,22]. In recent years, with the development of surface plasmon optics, surface enhanced Raman scattering (SERS) spectroscopy has attracted increasing attention. Due to the interaction between the excitation light and the noble metal nanomaterials, surface plasmon resonance (SPR) is generated, which leads to electromagnetic field enhancement at “hot spots” and amplifies the Raman signal by 10^6^–10^9^ times. Thus, the SERS detection sensitivity can reach a single molecular level [23,24]. In addition, SERS inherently has the advantages of a short response time, no interference from an aqueous solution, low sample pretreatment requirement, high accuracy, and no damage to the samples [25,26,27,28,29,30,31,32]. Therefore, combining SERS with IFA is a promising method and has been previously reported for rapid and highly sensitive detection of multiplex prostate cancer biomarkers and inflammatory biomarkers in human serum [33,34]. However, the study of this method for the determination of IL-6 in dairy cow milk is still barely reported.

In this study, a gold-silver core shell SERS nanotag with 4-MBA as Raman molecule was prepared and used for labelling in IFA for highly sensitive and quantitative detection of IL-6 in milk. As shown in Figure 1a, first, the surface of colloidal gold was modified with 4-MBA, and then the silver shell was wrapped outside to rationally design the Au^4-MBA^@Ag SERS nanotag. Then, it was conjugated with an Anti-IL-6 labeled antibody to form a bio-functionalized SERS nanotag. After that, the bio-functionalized SERS nanotag was mixed with the target IL-6 and dropped onto the test region of the kit, and maintained for 5 min for an immune reaction. Subsequently, it was washed and finally the SERS signal of the test spot was detected, which is schematically depicted in Figure 1b. Due to the high surface to volume ratio (SVR) of the detection membrane and the high sensitivity of the SERS nanotag, a satisfactory result with a wide linear range and low detection limit was obtained. It was proven that the proposed SERS IFA method was reliable and sensitive for the detection of IL-6 in milk.

## 2. Materials and Methods

### 2.1. Materials and Instruments 

Chloroauric acid (HAuCl_4_·3H_2_O), silver nitrate (AgNO_3_), ascorbic acid (AA) and trisodium citrate (Na_3_C_6_H_5_O_7_) were provided by Sigma-Aldrich (St. Louis, MO, USA). Bovine serum albumin (BSA), phosphate buffered saline (PBS, 0.1 M, and pH = 7.4), Tween 20, and 4-mercaptobenzoic acid (4-MBA) were obtained from Alfa-Aesar (Shanghai, China). Recombinant IL-6 and mouse anti-human IL-6 monoclonal antibody were commercially purchased from Beijing Key-bio Biotech Co., Ltd. (Beijing, China). IL-8, IL-10, CRP (C-reactive protein), and SAA (Serum amyloid A) were purchased from Biodee Biotechnology Co., Ltd. (Shanghai, China). The IFA components, including the nitrocellulose membranes, and absorbent pad were purchased from EMD Millipore Co., Ltd. (Billerica, MA, USA). Glassware was first soaked in a freshly prepared aqua regia solution (HCl/HNO_3_) overnight, and then rinsed with water before use. All distilled water was purified to Millipore Milli-Q quality. All of the chemicals were used as received.

SERS measurements were performed using a confocal Raman microscope IDSpec Aurora J300 and the objective magnification was 20× with a numerical aperture of 0.5. Its operating wavelength was 633 nm with a laser power of 10 mW, and providing a high-resolution Raman scattering spectrum with a resolution of 1 cm^−1^. The absorption spectrum of the nanoparticles was measured using a DU800 spectrometer and its scanning wavelength was from 200 to 800 nm. Scanning electron microscope (SEM) photographs were taken on a Mira3LMH (Tescan, Shanghai, China) with an accelerating voltage of 10 kV to characterize Au nanoparticles and SERS nanotags. The zeta potential and size distribution were acquired by a Zetasizer Nano Z (Malvern Instruments Ltd., Westborough, MA, USA). Moreover, the structure of the SERS nanotag was characterized using a transmission electron microscope (Hitachi 7650B, Berkshire, UK) with a resolution of 0.2 nm at an accelerating voltage of 40 kV.

### 2.2. Synthesis of Au^4-MBA^@Ag SERS Nanotags

The Au^4-MBA^@Ag were synthesized according to our previous paper [34], and the preparation procedure is illustrated in Figure 1a. Firstly, colloidal Au NPs were prepared using a method of reduction of chloroauric acid with sodium citrate according to the well-established Frens method, with minor modifications [35]. Briefly, 200 mL of chloroauric acid aqueous solution (0.01%, *w*/*v*) was heated under condensate reflux with magnetic stirring. When the solution was boiled, 1.5 mL of sodium citrate solution (1.0%, *w*/*v*) was quickly added once. After 10 min of reaction, the solution color changed to red wine, and was cooled down to room temperature. Then, the Au NPs were used as templates for the synthesis of Au^4-MBA^@Ag nanotags. Accordingly, 10 µM 4-MBA ethanol solution was added to 20 mL AuNP solution and vigorously stirred for 30 min. Then, 4-MBA molecular was connected to the surface of the AuNPs by the formation of Au-S bonds. Subsequently, 1 mL of 0.1 M ascorbic acid was added to Au^4-MBA^ solution, and then 200 µL of 2.5 mM AgNO_3_ was added dropwise under vigorous stirring. Because of the reduction of AgNO_3_ by ascorbic acid, the Ag shell was gradually surrounded by Au^4-MBA^ NPs, and eventually formed the Au^4-MBA^@Ag SERS nanotags. The Ag shell thickness was optimized to achieve maximum Raman enhancement [36].

### 2.3. Conjugation of IL-6 Antibody with Au^4-MBA^@Ag

Anti-IL-6 antibody labelled Au^4-MBA^@Ag nanotags were prepared as follows. Firstly, different volumes of K_2_CO_3_ (0.1 M) and HCl (0.1 M) were added to the prepared Au^4-MBA^@Ag solution (10 mL) to adjust the pH value to 8.5. After 3 min, 20 μL of labelled mAb against IL-6 (1.0 mg mL^−1^) was added to the Au^4-MBA^@Ag solution at 25 °C for 1 h. Subsequently, 200 µL of 5% BSA was added to the Au^4-MBA^@Ag conjugates and was allowed to react for another 30 min to block the unspecific binding sites on the sites of the NPs. Finally, the resulting solution was centrifuged at 6000 rpm for 10 min at 4 °C and the supernatant was discarded to remove the unbound antibody and chemicals. The residues containing the IL-6 conjugated Au^4-MBA^@Ag SERS nanotag was diluted with pure water to 1 mL and stored at 4 °C before use [37,38].

### 2.4. Preparation of IL-6 Solutions

IL-6 powder was dissolved in 10 mM PBS solution to prepare an IL-6 sample with different concentrations, and the solutions were stored at 4 °C before use.

### 2.5. Detection of IL-6 in Milk

Bovine raw milk was produced by lactating Holstein cows at the cow farm of the Institute of Animal Science, Chinese Academy of Agricultural Sciences. Then, IL-6 powder (1.0 mg) was spiked into 100 mL of liquid milk to reach a concentration of 0.1 mg mL^−1^ IL-6 as an initial solution. Thereafter, different concentrations of 0.01, 0.5, 2, 50, and 200 ng mL^−1^ of IL-6 in milk were prepared by serial dilutions of the initial solution. In order to prepare the milk samples for SERS detection, first, 4 mL of acetonitrile (acetone or ethyl alcohol) was added to 4 mL of the spiked milk sample [39]. Subsequently, the mixture was dispersed by ultrasonic for 5 min before centrifugation at 6000 rpm for 10 min. After that, 2 mL of deionized water was added into the centrifuge tube with the precipitate, and then the mixture was dispersed by ultrasonic for another 5 min and centrifuged at 6000 rpm for 5 min. Finally, the supernatant was transferred to a clean tube for measurement. The milk sample without IL-6 was used as the control.

### 2.6. Data Analysis

Every test was performed five times for repeated measurements in the current study. A smart phone equipped with a high-resolution camera was used to acquire visual pictures from NC membrane of this IFA kit within 5 min after each measurement. The images were processed using ImageJ software. The measured Raman spectra and visual intensity values were analyzed by Origin 2019 (OriginLab, Hampton, MA, USA). All the acquired data were shown as mean values with standard deviations.

### 2.7. The Principle of the SERS IFA Method

In this study, a tube was employed to premix the sample solution and the detection reagent, and was then subjected to the IFA kit. Compared with normal two-step titration, this assay format provided a homogeneous system for antigen-antibody binding, which could produce a higher sensitivity [40,41]. The principle of the SERS-based IFA for IL-6 was similar to that based on traditional colloidal NPs, and the scheme is illustrated in Figure 1b. The IL-6 mAb antibodies were immobilized on the test region of the nitrocellulose membrane by physical adsorption and were used as the capture reagent, while the labeled IL-6 mAb antibody conjugated Au^4-MBA^@Ag SERS nanotags (Au^4-MBA^@Ag@Ab) were used as the detection reagent. In a positive situation (the sample contained IL-6 target and the concentration was high), the Au^4-MBA^@Ag@Ab was specifically bound with IL-6, and then dropped on the IFA kit. The solution was infiltrated through the NC membrane by capillary force and was captured by coating antibodies, forming sandwich immune complexes in the test region. After that, washing three times was done to reduce the nonspecific adsorption of the nanotags on the NC membrane. In this case, due to the accumulation of the immune complexes, a brown-yellow dot would appear on the NC membrane. In a negative situation (the sample without IL-6 target), the sample could not bind with Au^4-MBA^@Ag@Ab. When applied in the test region of the NC membrane, they could not be captured by the coating antibody, leading to no immune complexes being captured in the test region and no color change. Therefore, the degree of color intensity as well as the SERS signal on the test zone was positively related to the concentration of IL-6 in the sample solution. Moreover, the SERS intensity that arose from Au^4-MBA^@Ag captured on the test zone was measured with Raman equipment, allowing for more sensitive and extensive measurement than the conventional colorimetric method.

## 3. Results

### 3.1. Characterization of Au^4-MBA^@Ag SERS Nanotag

In this study, an Au^4-MBA^@Ag SERS nanotag was adopted for the quantitative analysis of IL-6 in milk. Compared with the other protocols, it had merits such as being easy to use, and had a fast procedure and high sensitivity. AuNP cores were prepared by citrate reduction of the chloroauric acid solution, as described above [42]. The Figure 2a inset displays the colloid gold solution in purple, indicating the AuNPs were within 35–60 nm. Furthermore, most AuNPs had a round or spherical shape with a good dispersibility in the TEM image, as shown in Figure 2a. Based on the statistical analysis of the 100 AuNPs, the size distribution information is shown in Figure 2c. The average particle diameter was 55 ± 5 nm with a maximum absorption peak at 535 nm (Figure 3d). Subsequently, the Raman molecule 4-MBA was added to 20 mL Au colloids, which could be easily attached to the AuNP surface by the Au-S bond. Generally, the more 4-MBA molecules embedded into the gap of bimetal, the higher the SERS signal that will be generated, due to the ultrahigh enhancement of the electromagnetic field confined in the gap [43]. However, when excess 4-MBA was added to Au colloids, they would neutralize the surface change of AuNP and decrease its monodispersity, leading the AuNPs to aggregate or even precipitate, which would greatly affect the SERS intensity. To determine the proper amount of 4-MBA for the best SERS performance, different volumes of 4-MBA with a concentration of 10 µM were mixed with 20 mL of Au colloids. The averaged Raman spectra are shown in Figure 3a. Two dominated Raman peaks at 1076 and 1585 cm^−1^ were observed for Au^4-MBA^@Ag, which were attributed to the in-plane ring breathing mode coupled with the *ν*(C-S) and aromatic ring *ν*(C-C) vibration mode of 4-MBA molecular, respectively [44]. Since the Raman intensity at 1076 cm^−1^ was higher than that of 1585 cm^−1^, it was used as the characterized Raman peak for the quantitative analysis. Furthermore, with different amounts of 4-MBA, the SERS intensities at 1076 cm^−1^ are shown in Figure 3b. Alone, with the increasing amount of 4-MBA, the Raman intensity was significantly increased and then decreased, and the highest SERS intensity appeared when the volume of 4-MBA was 90 µL. Moreover, after the modification of 4-MBA, the UV–VIS peak of the Au NPs slightly red-shifted to 538 nm (Figure 3d), indicating that 4-MBA was successfully anchored on the Au surface. Thus, this amount of 4-MBA was optimal for the preparation of stable Au^4-MBA^ and was chosen for the next step.

For the preparation of Au^4-MBA^@Ag SERS nanotags, ascorbic was acted as a reductive agent mixed with Au^4-MBA^ colloid. Thereafter, AgNO_3_ was added dropwise under magnetic stirring. As a result, the Ag (+) in the AgNO_3_ solution was reduced to Ag (0) and was selectively grown on the Au cores. Owing to the match of the crystalline lattice between Ag and Au, the core–shell Au^4-MBA^@Ag SERS nanotags were formed [45]. As the thickness of Ag shell had a significant effect on SERS signal intensity, therefore the amount of AgNO_3_ was investigated. Figure 3c shows the SERS intensity at 1076 cm^−1^ of Au^4-MBA^@Ag with different amounts of AgNO_3_. It can be seen that Au^4-MBA^@Ag exhibited a higher SERS signal intensity than pure Au^4-MBA^. When the amount of AgNO_3_ (2.5 mM) increased to 0.9 mL, the highest Raman intensity of Au^4-MBA^@Ag was obtained. Thus, 0.9 mL of AgNO_3_ was chosen as the optimum volume for the preparation of Au^4-MBA^@Ag SERS nanotags. Moreover, as shown in Figure 3d, the main UV–VIS peak shifted from 538 nm to 515 nm after the coating of the Ag shell. Meanwhile, an Ag shoulder appeared at 400 nm, confirming the formation of Au^4-MBA^@Ag NPs. The picture of the Au^4-MBA^@Ag colloid is shown in the Figure 2b inset. Compared with the Au colloid displayed in the Figure 2a inset, an obvious orange yellow color appeared in the Au^4-MBA^@Ag colloid, indicating that Ag was successfully grown on the surface of Au^4-MBA^ nanoparticles. The TEM image of Au^4-MBA^@Ag NPs is shown in Figure 2b. It was clear that the NPs had a uniform particle size and good mono-dispersity. Meanwhile, the size distribution shown in Figure 2d reveals that the diameter of Au^4-MBA^@Ag was in the range of 58 ± 3 nm. Compared to the TEM image and size distribution of Au NPs in Figure 2a,c, it could be deduced that the average Ag shell thickness of the Au^4-MBA^@Ag core shell NPs was about 3 nm.

### 3.2. Optimization of the Assay Conditions

In this SERS IFA, it is well known that the amount of coating antibodies, the amount of antibodies used for preparing of Au^4-MBA^@Ag@Ab, the volume of labelled SERS nanotags, and the washing times were crucial for improving the detection sensitivity. To evaluate the optimized condition, B_0.1_/B_0_ was introduced to decrease the interface of background and was used as an assessing criterion. B_0.1_ and B_0_ were the Raman intensities at 1076 cm^−1^, with concentrations of IL-6 of 0.1 and 0 ng mL^−1^, respectively. The bigger the B_0.1_/B_0_ value, the higher the sensitivity of the assay [46]. To choose the optimum concentration of the labelled IL-6 antibody, different amounts of labelled antibodies (from 5 to 30 μL) with a concentration of 1 mg mL^−1^ were successively added to 5.0 mL of Au^4-MBA^@Ag NPs. Then, the B_0.1_/B_0_ value was investigated (Figure 4a). It was observed that with the increase in antibodies, the B_0.1_/B_0_ value increased and peaked at 20 μL, and then decreased with further additions. The reason for this could be that the redundant free antibodies in the solution may reduce the amount of Au^4-MBA^@Ag@Ab bound on the NC membrane, leading to declining B_0.1_/B_0_ values. Therefore, the optimal amount of antibody for preparing Au^4-MBA^@Ag@Ab was 20 μL. Similarly, when different volumes (from 2 to 10 μL) of Au^4-MBA^@Ag@Ab NPs were used to mix with IL-6, and 3 μL of coating IL-6 antibody with concentrations from 0.5 to 3 mg mL^−1^ were coated on the test dot. As shown in Figure 4b,c, 6 μL of Au^4-MBA^@Ag@Ab and 2.0 mg mL^−1^ coating IL-6 antibody were considered as the optimal assay conditions. In addition, in order to further improve the sensitivity and reduce the nonspecific immunoreaction on the NC membrane, the washing times were also examined. As shown in Figure 4d, with the increase in washing times, the B_0.1_/B_0_ value sharply decreased and then remained stable after washing three times. Thus, washing three times occurred in the immunoassay. These optimized experimental conditions could lay the foundation for the quantitative analysis of IL-6 in the next experiment.

### 3.3. Quantitative SERS Measurements of IL-6

Under the optimized conditions, the feasibility of the SERS improved IFA was tested. First, analysis of IL-6 with concentrations of 100 and 0 ng mL^−1^ was performed. As expected, with the sample containing IL-6, the target could combine with SERS nanoprobes and was captured by the coating antibodies, and formed a sandwich immune complex. Meanwhile, with the accumulation of the immune complexes, there will be color change on the test dot to identify the existence of IL-6 (Appendix A). As shown in the SEM image of Appendix A, there were bright small dots on the fiber pores of the test NC membrane, and this observation confirmed the accumulation of the sandwich immune complexes. In contrast, with the absence of IL-6, there was no obvious bright spot on the SEM image, and no color change at the test spot and no obvious SERS signal could be detected (Appendix A). These results showed that IL-6 could be quantitatively detected by this proposed biosensor. Then, the proposed SERS IFA platform was used for the quantitative analysis of IL-6 at concentrations of 0, 2 × 10^−5^, 2 × 10^−4^, 2 × 10^−3^, 2 × 10^−2^, 0.2, 2.0, 20, and 200 ng mL^−1^. They were prepared by diluting the stock solution with the PBS solution. Briefly, 10 μL of the IL-6 antigen of each concentration was blended with Au^4-MBA^@Ag@Ab and then applied to the immune-kit and maintained for 5 min. In this process, it could be seen that the mixture gradually migrated downward through the NC membrane by the capillary force of the absorption pad, and a color dot appeared. 

As shown in Figure 5a, with the decreased concentration of IL-6, the intensity of color of the test dot gradually weakened, and even disappeared when concentration less than 2.0 pg mL^−1^. Thus, the colorimetric assay worked only in the range of 200 ng mL^−1^–2 pg mL^−1^, and the results could only be semi-quantitative analyzed [47]. In order to obtain the highly sensitive and accurate detection of IL-6 for the judgement of mastitis in dairy cows, the SERS intensity of the test dot was measured using a Raman spectrometer. The average SERS spectra are shown in Figure 5b. The SERS intensity at 1076 cm^−1^ concomitantly increased with the increase in IL-6 concentrations. Based on this result, a standard curve in the form of the logarithmic concentration of IL-6 versus SERS peak intensity at 1076 cm^−1^ was established (Figure 5c). As shown in the Figure 5c inset, the linear dynamic range (LDR) spanned five orders of magnitude. The linear relationship between the SERS intensity at 1076 cm^−1^ and the logarithmic IL-6 concentration was y=4461.2x+21,908.5 with *R*^2^ = 0.987. On the basis of three times the standard deviation of the blank sample, the limit of detection (LOD) of IL-6 was calculated as 0.074 pg mL^−1^, which was lower than those recently reported (Appendix A). This value was also much lower than the threshold of the clinical mastitis of dairy cows, demonstrating the ultra-high sensitivity of the proposed SERS IFA for IL-6 detection.

### 3.4. Reproducibility of the SERS IFA

In order to investigate the reproducibility of SERS IFA, the Raman intensity at 1076 cm^−1^ of ten different points in the center region of the test dot were examined with the concentrations of IL-6 solution at 10, 1.0, 0.1, and 0.01 ng mL^−1^, respectively. As shown in Figure 6, for each concentration, the relative standard deviation (RSD) value of the SERS intensity acquired from those ten different points were 4.57%, 5.32%, 3.87%, and 6.72%, respectively, indicating a high reproducibility and precision of the proposed SERS method.

### 3.5. Specificity of the SERS IFA

Generally, the cross-reaction usually affected the test results, and was crucial for the assessment of the specificity and selectivity of the immunoassay. To investigate the cross-reaction of this SERS IFA biosensor, SERS detection of other cytokines or inflammatory biomarkers including IL-8, IL-10, CRP, and SAA were performed. All of the compounds including IL-6 were prepared with a concentration of 100 pg mL^−1^, and were applied in SERS IFA procedures. Typical SERS spectra and Raman intensities recorded in response to these compounds are shown in Figure 7a,b. Compared with the signal of IL-6, the signal of the other compounds was neglectable, indicating there was no cross interference between IL-6 and the other compounds. These results could be attributed to two major reasons. The first one was that the utilized monoclonal capture and labelled antibodies had a high specificity with IL-6. The other one was that this platform features easy washing, noninterfering nature of the components, and high sensitivity. Therefore, this proposed bioassay system shows great potential for cow mastitis diagnosis.

### 3.6. Detection of IL-6 in Spiked Milk Sample

After pretreatment, the milk sample was used to estimate the accuracy and practicality of the SERS IFA. Firstly, Western blot analysis was performed to confirm that there was no IL-6 protein marker in the milk sample (Figure 8a). Subsequently, different concentrations of IL-6 were added to bovine whey to obtain a mixture with concentrations from 1 pg mL^−1^ to 1 µg mL^−1^. Quantitative analysis of IL-6 was carried out by measuring the SERS signal intensity of the test dot. As shown in Figure 8b, the Raman intensity increased with the increase in IL-6 concentrations. Based on the Raman signal intensity of 4-MBA, a linear calibration plot was constructed for the quantitative evaluation of IL-6 (Figure 8c). There was a significantly linear relationship between the intensity of the Raman signal and the logarithm of the IL-6 concentration (Figure 8c). The regression equation was y=3688.8x+10,380.9 with *R*^2^ = 0.991, and the minimum detection limit was 0.35 pg mL^−1^, which indicated that the proposed biosensor had a high detection accuracy for IL-6 in milk. At the same time, different concentrations of IL-6, ranging from 0.18 to 504.1 ng mL^−1^, were added to the milk sample, and were determined with the electrochemiluminescence assay (ECLIA) kit and the proposed method (Appendix A), respectively. The comparison of the two methods is shown in Figure 8d. The correlation coefficient between the methods was close to 1.0. Moreover, the average recovery of IL-6 calculated from the spiked milk sample was 87.0% to 102.7%, with RSD values ranged from 2.5% to 8.7% (Appendix A). These results suggest that the high accuracy of the SERS IFA was in good consistency with ECLIA. In addition, the sample consumption was less and the operation procedure was more convenient in comparison with ECLIA. These advantages strongly illustrated that the proposed method could provide a potential practical application in the early diagnosis of mastitis for dairy cows.

## 4. Conclusions

In this study, a novel SERS-based IFA using the Au^4-MBA^@Ag conjugated antibody for the quantitative and sensitive detection of IL-6 in milk was developed. Raman peak intensity centered at 1074 cm^−1^ was monitored and its variation was used to evaluate the content of IL-6. Under optimal conditions, a logistic relationship between the Raman intensity, and the logged IL-6 concentration was obtained in the range from 2 × 10^−5^ to 200 pg mL^−1^. The LODs calculated from three standard deviations were 0.074 and 0.35 pg mL^−1^ for IL-6 in the PBS and milk samples, respectively. Meanwhile, the determination of IL-6 was also performed by ECLIA and SERS IFA in the spiked milk sample. The recovery of the spiking experiment was 87.0–102.7%, with coefficients of variation that ranged from 2.5% to 8.7%, demonstrating the high accuracy of the proposed method. Additionally, compared with the ECLIA method, SERS IFA had the advantages of simple operation, time saving, and good reproducibility. This proposed method also has the advantages of being low cost and having a high precision compared with traditional dairy cow mastitis detection laws, such as microbiological diagnosis, somatic cell count, and nuclear drug sensitivity tests. Encouragingly, SERS IFA equipped with a portable Raman spectrometer developed in the current study could potentially provide a practical and effective method for early milk IL-6 detection in cow mastitis diagnosis.

## Figures and Tables

**Figure 1 nanomaterials-12-01091-f001:**
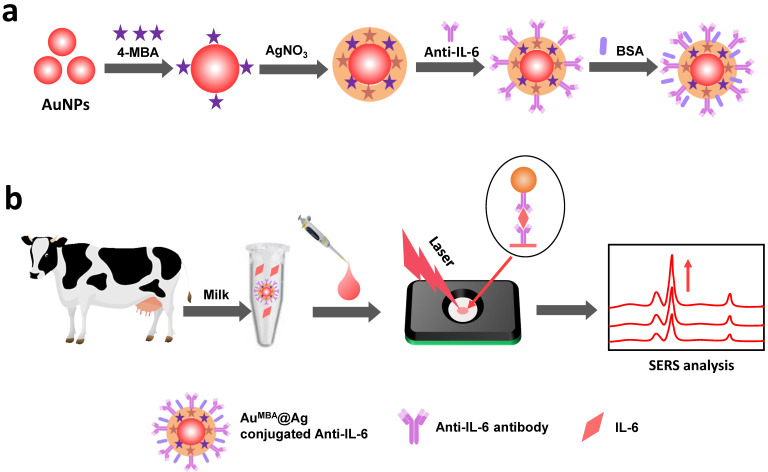
Schematic illustration of (**a**) the synthesis of bio-functionalized core-shell SERS nanotag; (**b**) the detection of IL-6 by core-shell SERS nanotags combined with the IFA kit.

**Figure 2 nanomaterials-12-01091-f002:**
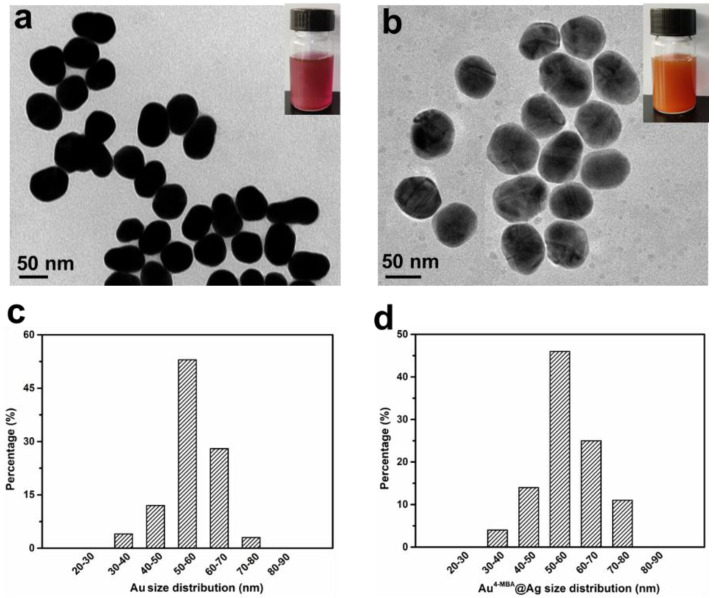
(**a**–**c**) TEM image and size distribution of Au NPs; (**b**–**d**) TEM image and size distribution of Au^4-MBA^@Ag. Insets of (**a**,**b**) are the photographs of their corresponding colloidal dispersion.

**Figure 3 nanomaterials-12-01091-f003:**
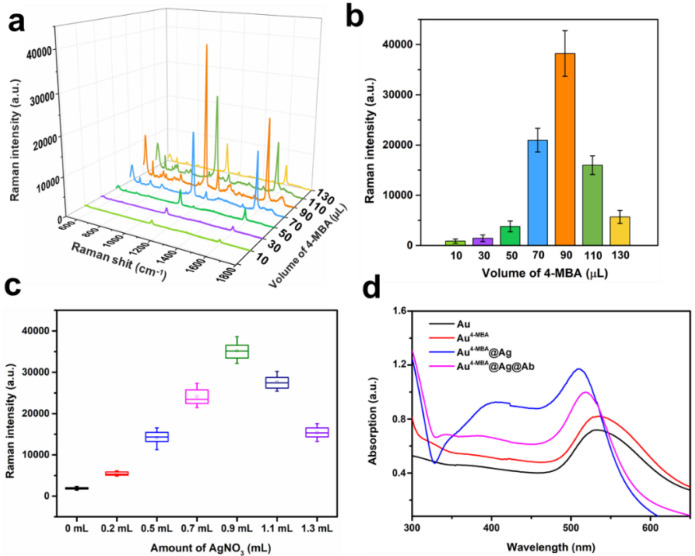
(**a**) Raman spectra and (**b**) Raman intensities at 1076 cm^−1^ of different amounts of 4-MBA; (**c**) Raman intensities at 1076 cm^−1^ of different amounts of AgNO_3_; (**d**) UV-VIS extinction spectra of Au, Au^4-MBA^, Au^4-MBA^@Ag, and Au^4-MBA^@Ag@Ab.

**Figure 4 nanomaterials-12-01091-f004:**
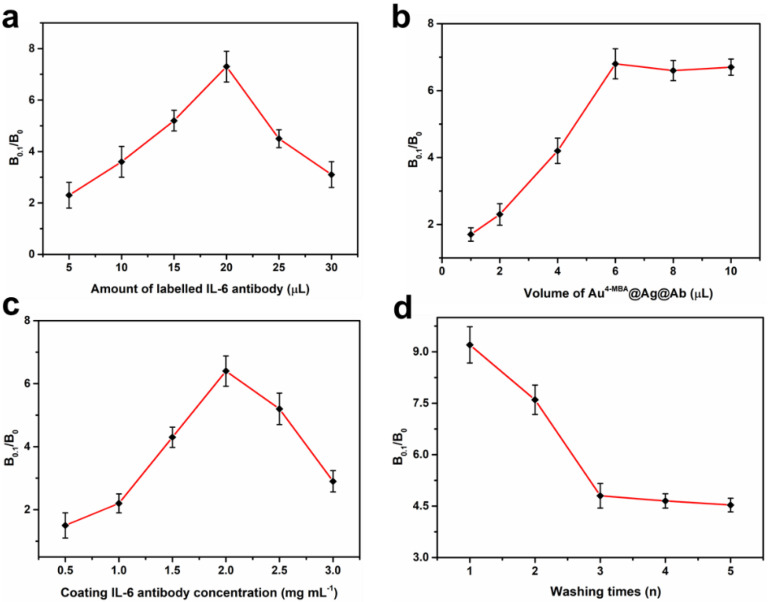
Optimization of the SERS IFA conditions: (**a**) optimization of the amount of labelled antibody in the preparation of Au^4-MBA^@Ag@Ab; (**b**) optimization of the amount of Au^4-MBA^@Ag@Ab solution; (**c**) optimization of coating IL-6 antibody concentration on the NC membrane; (**d**) optimization of different washing times.

**Figure 5 nanomaterials-12-01091-f005:**
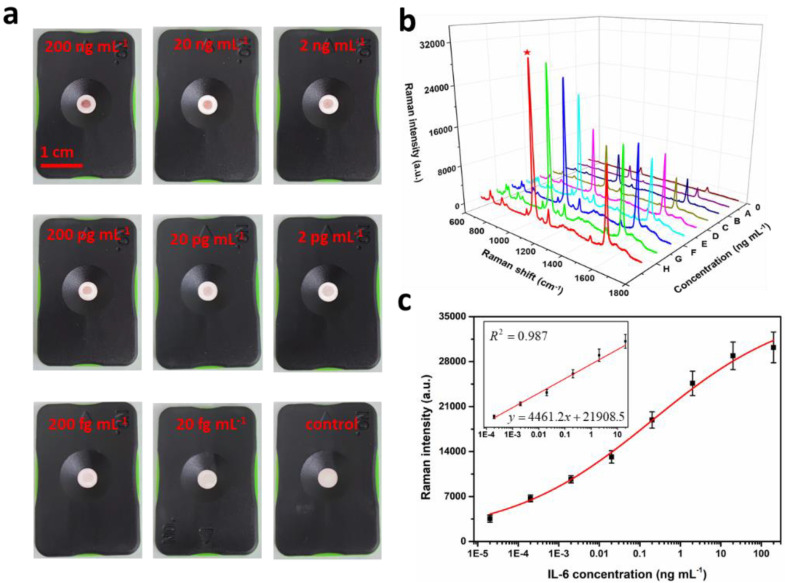
(**a**) Image of the test dot (4 mm in diameter) for different concentrations of IL-6; (**b**) averaged Raman spectra of test dot for different concentrations of IL-6 (A-H corresponding to 20 fg mL^〓1^-200 ng mL^−1^); (**c**) the calibration curve of IL-6, and the inset shown the linear part. The error bars are calculated from three measurements.

**Figure 6 nanomaterials-12-01091-f006:**
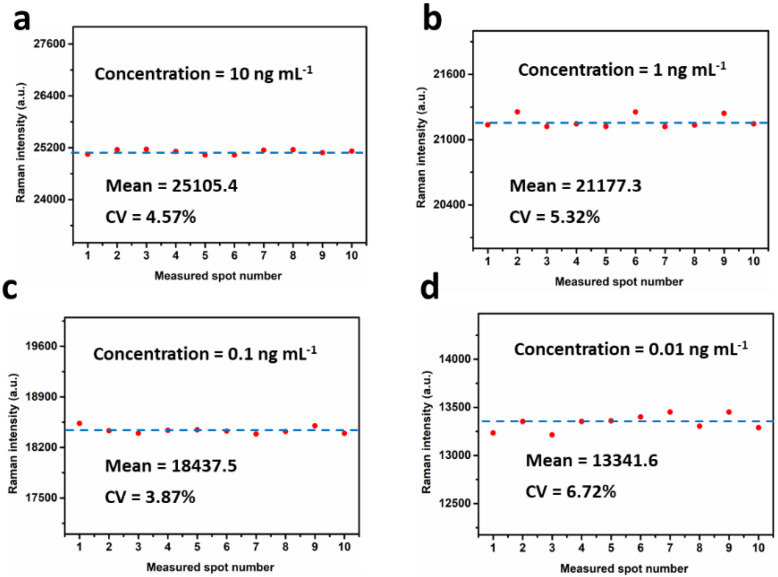
The SERS intensity of Au^4-MBA^@Ag at 1076 cm^−1^ from ten different spots in the center part of the test spot with concentrations of (**a**) 10 ng mL^−1^, (**b**) 1.0 ng mL^−1^, (**c**) 0.1 ng mL^−1^, and (**d**) 0.01 ng mL^−1^, respectively.

**Figure 7 nanomaterials-12-01091-f007:**
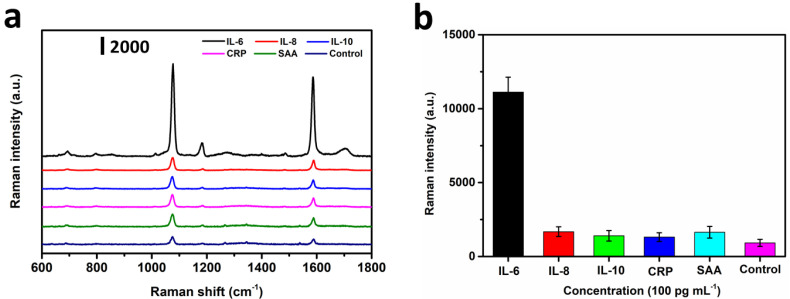
Specificity of SERS IFA. (**a**) SERS spectra of Au^4-MBA^@Ag at 1076 cm^−1^, and (**b**) Raman intensity for IL-6, IL-8, IL-10, CRP, and SAA at a concentration of 100 pg mL^−1^.

**Figure 8 nanomaterials-12-01091-f008:**
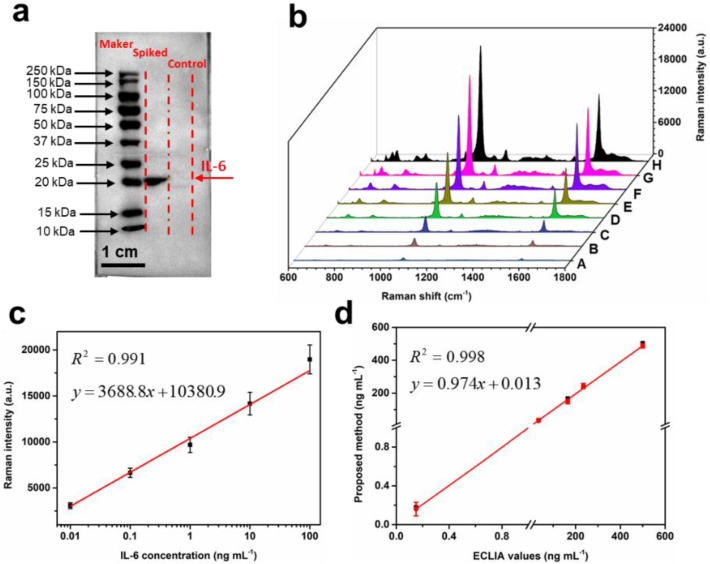
(**a**) Western blot analysis of IL-6 protein spiked in the milk; (**b**) SERS spectra of different concentrations (0 ng mL^–1^-1 µg mL^−1^) of IL-6 spiked in the milk and (**c**) the linear calibration curve; (**d**) comparison of detection results obtained from proposed method (red circles) and reference ECLIA method (black squares). Error bars are calculated from three measurements.

## Data Availability

Not applicable.

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
