# Peer review of "Quantitative Detection of Mastitis Factor IL-6 in Dairy Cow Using the SERS Improved Immunofiltration Assay"

_nanomaterials, 2022, doi:10.3390/nano12071091_

Round 1
Reviewer 1 Report
The authors describe the indirect SERS detection of interleukin-6 (IL-6) by using plasmonic nanoparticles formed by a gold core modified with 4-MBA label, recovered with a silver shell. Through immunoassay, IL-6 was adsorbed from spiked milk samples and the limit of detection at 0.35 pg mL-1 was obtained. The text is clear and the results are consistent, but some questions involving details in the procedure are presented below and must be better explained.
1- In p. 5, line 183, the authors state “The IL-6 mAb antibodies were immobilized on the test region of nitrocellulose membrane as the capture reagent”. Details must be provided on how the protein was immobilized in the NC membrane. In the sequence, at line 187 is affirmed “The solution infiltrated through NC membrane by capillary force and captured by coating antibodies”. However, none explanation is done about the washing of the NC membrane before taking SERS record, but nanotag may not have specific interaction with antibody and once that is soaking the substrate it can lead to the record of SERS of 4-MBA label. This explanation has to be done in the section 2.7.
2- If the required washing explanation, above cited, is related to the experiment presented in Fig. 4d, what is the reason by which the grater number of washing times do not lead to B0.1/B0 = 1?
Author Response
To Reviewer #1:
Comments: The authors describe the indirect SERS detection of interleukin-6 (IL-6) by using plasmonic nanoparticles formed by a gold core modified with 4-MBA label, recovered with a silver shell. Through immunoassay, IL-6 was adsorbed from spiked milk samples and the limit of detection at 0.35 pg mL-1 was obtained. The text is clear and the results are consistent, but some questions involving details in the procedure are presented below and must be better explained.
Question 1: In p. 5, line 183, the authors state “The IL-6 mAb antibodies were immobilized on the test region of nitrocellulose membrane as the capture reagent”. Details must be provided on how the protein was immobilized in the NC membrane. In the sequence, at line 187 is affirmed “The solution infiltrated through NC membrane by capillary force and captured by coating antibodies”. However, none explanation is done about the washing of the NC membrane before taking SERS record, but nanotag may not have specific interaction with antibody and once that is soaking the substrate it can lead to the record of SERS of 4-MBA label. This explanation has to be done in the section 2.7.
Answer: We thank for the reviewer’s valuable suggestion. â… . Because there are abundant hydroxyl hydrophilic groups on the surface of nitrocellulose membrane, when the coated antibody is dropped on the NC membrane, it can infiltrate the membrane. After the NC membrane is dry, the coated antibody can be fixed on it by physical adsorption. Accordingly, we have changed the sentence to “The IL-6 mAb antibodies were immobilized on the test region of nitrocellulose membrane by physical adsorption and used as the capture reagent”. in Line 185 and highlighted in the revised manuscript. â…¡: In order to make the experimental process clearer, washing should be explained. Accordingly, the sentence of “After that, three times of washing was done to reduce the nonspecific adsorption of nanotags on the NC membrane.” is added in Line 192 and highlighted in the revised manuscript.
Question 2: If the required washing explanation, above cited, is related to the experiment presented in Fig. 4d, what is the reason by which the grater number of washing times do not lead to B0.1/B0 = 1?
Answer: We thank for the reviewer’s good question. In theory, with the increase of washing times, the value of B0.1/B0 should be close to 1. However, in practical experiments, non-specific adsorption is inevitable. Although increasing the washing times could reduce the nonspecific adsorption, the result found that the B0.1/B0 value remained almost unchanged after three times of washing. Thus, three times of washing was used in the experiment.
List of Changes (Red in the context)
- All the modified graphics have been inserted in the revised manuscript.
- Words of “molecule” and “yuan” have been added on Page 1 in “Abstract” and “Introduction” section of the revised manuscript.
- Words of “glycoproteins” and “SPR”, and sentences of “These results are insufficient for effective diagnosis of inflammatory diseases in dairy cows. Therefore, it is urgent and challenging to establish a simple and easy method to quantitatively detect IL-6 in milk.” was added on Page 2.
- Words of “molecule” and sentence of “IL-8, IL-10, CRP (C-reactive protein), and SAA (Serum amyloid A) were purchased from Biodee Biotechnology Co., Ltd. (Shanghai, China).” , “and the objective magnification is 20X with the numerical aperture of 0.5” and “with laser power of 10 mW” has been added on Page 3 the revised manuscript.
- Words of “kV” has been rewritten on Page 4 of the revised manuscript.
- Words of “molecule” and sentence of “The IL-6 mAb antibodies were immobilized on the test region of nitrocellulose membrane by physical adsorption and used as the capture reagent as the capture reagent” and “After that, three times of washing was done to reduce the nonspecific adsorption of nanotags on the NC membrane.” have been added on Page 5 of the revised manuscript.
- Words of “antibody” has been added on Page 7 of the revised manuscript.
- Sentences of “IL-8, IL-10, CRP, and SAA” has been added on Page 10 of the revised manuscript.
- Sentences of “By the way, this proposed method also has the advantages of low cost and high precision compared with traditional dairy cow mastitis detection laws, such as microbiological diagnosis, somatic cell count and nuclear drug sensitivity test. Encouragingly,” and “National Key R&D Program of China (Grant No. 2021YFD2000804)” have been added on Page 12 of the revised manuscript.
- Words of “1150f2” has been added on Page 13 of the revised manuscript.
- Words of “Spectrochim. Acta. A Molec. Biomol. Spectrosc” has been rewritten on Page 14 of the revised manuscript.
- The grammar has been improved and some other minor modifications have been made in a careful proof reading.

Reviewer 2 Report
The Authors developed and tested a quantitative test for the detection of mastitis factor IL-6 in the dairy cow. They used SERS technique together with custom-made SERS-active nanostructures. The results demonstrate that the detection limit of IL-6 in milk was 0.35 pg mL-1, which was below the threshold value of 254.32 pg mL-1. The recovery of the spiking experiment was 87.0%-102.7% with coefficients of variation below 9.0%. That demonstrate the high accuracy and precision of the proposed assay.
The manuscript can be accepted for publication in Nanomaterials-especially for the Advanced Nanomaterials for Sensing Applications issue-after minor revision. The remarks are:
The manuscript should be re-read, preferably by someone with a very good command of English. There are some grammatical errors in the article (e.g. line 66, 69), or excessive use of some words (e.g. therefore, moreover). This is very important for the Abstract and Introduction: they must be easy to read and follow.
Figure 1 - scheme a should be improved. The direction (the flow) should be from left to right, not the opposite.
line 121 and 125: please use kV instead of Kv
Figure 2 - the histograms could be b/w (remove colors),
Figure 5 - the quality of this figure has to be improved, especially Fig 5a.
Please elaborate (especially in Conclusion) more on other methods of detection of mastitis, and highlight the advantages of the proposed method.
Author Response
To Reviewer #2:
Comments: The Authors developed and tested a quantitative test for the detection of mastitis factor IL-6 in the dairy cow. They used SERS technique together with custom-made SERS-active nanostructures. The results demonstrate that the detection limit of IL-6 in milk was 0.35 pg mL-1, which was below the threshold value of 254.32 pg mL-1. The recovery of the spiking experiment was 87.0%-102.7% with coefficients of variation below 9.0%. That demonstrate the high accuracy and precision of the proposed assay.
The manuscript can be accepted for publication in Nanomaterials-especially for the Advanced Nanomaterials for Sensing Applications issue-after minor revision. The remarks are:
Question 1: The manuscript should be re-read, preferably by someone with a very good command of English. There are some grammatical errors in the article (e.g. line 66, 69), or excessive use of some words (e.g. therefore, moreover). This is very important for the Abstract and Introduction: they must be easy to read and follow.
Answer: We thank for the reviewer’s careful reading and good suggestion. Accordingly, we have re-read the manuscript throughout, and reduced the usage of “therefore” and “moreover” in Abstract and Introduction section. We also rewritten the sentence as “These results are insufficient for effective diagnosis of inflammatory diseases in dairy cows. It is urgent and challenging to establish a simple and easy method to quantitatively detect IL-6 in milk.” in Line 66 and highlighted it in the revised manuscript.
Question 2: Figure 1 - scheme a should be improved. The direction (the flow) should be from left to right, not the opposite.
Answer: We thank for the reviewer’s good suggestion. Accordingly, we have changed the flow of Figure 1a in the revised manuscript.
Question 3: line 121 and 125: please use kV instead of Kv.
Answer: We thank for the reviewer’s good suggestion. Accordingly, we have changed “Kv” to “kV” in Line 122 and 127, and highlight them in the revised manuscript.
Question 4: Figure 2 - the histograms could be b/w (remove colors).
Answer: We thank for the reviewer’s good suggestion. Accordingly, we have changed the colors of Figure 2c and d in the revised manuscript.
Question 5: Figure 5 - the quality of this figure has to be improved, especially Fig 5a.
Answer: We thank for the reviewer’s good suggestion. Accordingly, we have improved the all the figures’ quality of Figure 5 in the revised manuscript.
Question 6: Please elaborate (especially in Conclusion) more on other methods of detection of mastitis, and highlight the advantages of the proposed method.
Answer: We thank for the reviewer’s valuable suggestion. Accordingly, we have added the sentence of “By the way, this proposed method also has the advantages of low cost and high precision compared with traditional dairy cow mastitis detection laws, such as microbiological diagnosis, somatic cell count and nuclear drug sensitivity test. Encouragingly,” at Line 408 and highlighted it in the revised manuscript.
List of Changes (Red in the context)
1. All the modified graphics have been inserted in the revised manuscript.
2. Words of “molecule” and “yuan” have been added on Page 1 in “Abstract” and “Introduction” section of the revised manuscript.
3. Words of “glycoproteins” and “SPR”, and sentences of “These results are insufficient for effective diagnosis of inflammatory diseases in dairy cows. Therefore, it is urgent and challenging to establish a simple and easy method to quantitatively detect IL-6 in milk.” was added on Page 2.
4. Words of “molecule” and sentence of “IL-8, IL-10, CRP (C-reactive protein), and SAA (Serum amyloid A) were purchased from Biodee Biotechnology Co., Ltd. (Shanghai, China).” , “and the objective magnification is 20X with the numerical aperture of 0.5” and “with laser power of 10 mW” has been added on Page 3 the revised manuscript.
5. Words of “kV” has been rewritten on Page 4 of the revised manuscript.
6. Words of “molecule” and sentence of “The IL-6 mAb antibodies were immobilized on the test region of nitrocellulose membrane by physical adsorption and used as the capture reagent as the capture reagent” and “After that, three times of washing was done to reduce the nonspecific adsorption of nanotags on the NC membrane.” have been added on Page 5 of the revised manuscript.
7. Words of “antibody” has been added on Page 7 of the revised manuscript.
8. Sentences of “IL-8, IL-10, CRP, and SAA” has been added on Page 10 of the revised manuscript.
9. Sentences of “By the way, this proposed method also has the advantages of low cost and high precision compared with traditional dairy cow mastitis detection laws, such as microbiological diagnosis, somatic cell count and nuclear drug sensitivity test. Encouragingly,” and “National Key R&D Program of China (Grant No. 2021YFD2000804)” have been added on Page 12 of the revised manuscript.
10. Words of “1150f2” has been added on Page 13 of the revised manuscript.
11. Words of “Spectrochim. Acta. A Molec. Biomol. Spectrosc” has been rewritten on Page 14 of the revised manuscript.
12. The grammar has been improved and some other minor modifications have been made in a careful proof reading.

Reviewer 3 Report
The submitted manuscript "Quantitative detection of mastitis factor IL-6 in dairy cow using SERS improved immunofiltration assay" reported a SERS-IFA based IL-6 test method for milk samples. The proposed method exhibited good sensitivity and specificity. Therefore, it is my opinion that a minor revision is necessary before its publication for Nanomaterials. Some comments for this manuscript are list below:
- The SERS-based method was proposed for cow IL-6. However, in the preparation of probes, the antibodies is anti-human IL-6. Why is anti-human antibody used?
- Why was the preparation of probes presented from right to left?
- What is the initial IL-6 concentration in the raw milk before spiking? The concentrations of IL-6 stated the concentration of spiked IL-6. Consider the origin IL-6 in the raw milk, is the concentration of total IL-6 higher than the stated concentration?
- There is no statement on how the measurement in milk was performed in the Materials and Methods section.
- What are the resources of IL-8, IL-10, CRP, SAA and PCT?
- It was listed “Mouse IgG, human IgG, anti-mouse IgG, and anti-human IgG” in the Materials and Methods section, but their usages are not found the manuscript.
- What are the results in Figure 4c about? In link 278 it says “coating IL-6” while in Figure 4c it says “coating IL-6 antibody”.
- The text should be carefully checked. Some found problems are:
pg mL-1 should be pg•ml.
b. in line 35, cost is nearly 200 million “what” every year .
c. line 47, “glycoprotein” should be “glycoproteins”.
d. in line 77, surface plasmon resonance is SPR, not LSPR).
e. in line 121, “10 Kv” should be “10 kV”.
Author Response
To Reviewer #3:
Comments: The submitted manuscript "Quantitative detection of mastitis factor IL-6 in dairy cow using SERS improved immunofiltration assay" reported a SERS-IFA based IL-6 test method for milk samples. The proposed method exhibited good sensitivity and specificity. Therefore, it is my opinion that a minor revision is necessary before its publication for Nanomaterials. Some comments for this manuscript are list below:
Question 1: The SERS-based method was proposed for cow IL-6. However, in the preparation of probes, the antibodies is anti-human IL-6. Why is anti-human antibody used?
Answer: We thank for the reviewer’s good question. Originally, we plan to use the anti-bovine antibody to prepare the detection probe, but the reagent was not available on the market. So, we replaced it with the anti-human antibody. Also, the results shows that IL-6 protein is highly conserved across species.
Question 2: Why was the preparation of probes presented from right to left?
Answer: We thank for the reviewer’s good question. To make the graph look more comfortable, we have changed the flow of Figure 1a in the revised manuscript.
Question 3:What is the initial IL-6 concentration in the raw milk before spiking? The concentrations of IL-6 stated the concentration of spiked IL-6. Consider the origin IL-6 in the raw milk, is the concentration of total IL-6 higher than the stated concentration?
Answer: We thank for the reviewer’s good question. In fact, before the spiking experiment, we performed western blot analysis to confirm that there was no IL-6 protein marker existed in the milk sample. Then, different concentrations of IL-6 were added to the blank milk sample. Therefore, the total concentration of IL-6 in the sample is the spiking concentration.
Question 4: There is no statement on how the measurement in milk was performed in the Materials and Methods section.
Answer: We thank for the reviewer’s good question. Actually, we have described the “Detection of IL-6 in milk” in Line 159 of the manuscript.
Question 5: What are the resources of IL-8, IL-10, CRP, SAA and PCT?
Answer: We thank for the reviewer’s good question. Their host animal is mouse.
Question 6: It was listed “Mouse IgG, human IgG, anti-mouse IgG, and anti-human IgG” in the Materials and Methods section, but their usages are not found the manuscript.
Answer: We thank for the reviewer’s good suggestion and careful reading, and sorry for my wrong writing. Accordingly, we have rewritten the sentences of “IL-8, IL-10, CRP (C-reactive protein), and SAA (Serum amyloid A) were purchased from Biodee Biotechnology Co., Ltd. (Shanghai, China).” and “IL-8, IL-10, CRP, and SAA” in Line 109 and 354 and highlighted them in the revised manuscript.
Question 7: What are the results in Figure 4c about? In link 278 it says “coating IL-6” while in Figure 4c it says “coating IL-6 antibody”.
Answer: We thank for the reviewer’s good question and careful reading. In order to optimize the concentration of coating antibody, Figure c represents 3 μL of coating IL-6 antibody with different concentrations coated on the test dot. Meanwhile, we have added “antibody” in Line 282 and 284, and highlighted them in the revised manuscript.
Question 8: The text should be carefully checked. Some found problems are:
pg mL-1 should be pg•ml.
b. in line 35, cost is nearly 200 million “what” every year .
c. line 47, “glycoprotein” should be “glycoproteins”.
d. in line 77, surface plasmon resonance is SPR, not LSPR).
e. in line 121, “10 Kv” should be “10 kV”.
Answer: We thank for the reviewer’s good question and careful reading. We think pg mL-1 is equal to
pg•mL-1, and it is commonly used in the papers. Also, we have added “yuan”, “glycoproteins”, “SPR” and “kV” in Line 36, 47, 77 and 122, respectively, and highlighted them in the revised manuscript.
List of Changes (Red in the context)
1. All the modified graphics have been inserted in the revised manuscript.
2. Words of “molecule” and “yuan” have been added on Page 1 in “Abstract” and “Introduction” section of the revised manuscript.
3. Words of “glycoproteins” and “SPR”, and sentences of “These results are insufficient for effective diagnosis of inflammatory diseases in dairy cows. Therefore, it is urgent and challenging to establish a simple and easy method to quantitatively detect IL-6 in milk.” was added on Page 2.
4. Words of “molecule” and sentence of “IL-8, IL-10, CRP (C-reactive protein), and SAA (Serum amyloid A) were purchased from Biodee Biotechnology Co., Ltd. (Shanghai, China).” , “and the objective magnification is 20X with the numerical aperture of 0.5” and “with laser power of 10 mW” has been added on Page 3 the revised manuscript.
5. Words of “kV” has been rewritten on Page 4 of the revised manuscript.
6. Words of “molecule” and sentence of “The IL-6 mAb antibodies were immobilized on the test region of nitrocellulose membrane by physical adsorption and used as the capture reagent as the capture reagent” and “After that, three times of washing was done to reduce the nonspecific adsorption of nanotags on the NC membrane.” have been added on Page 5 of the revised manuscript.
7. Words of “antibody” has been added on Page 7 of the revised manuscript.
8. Sentences of “IL-8, IL-10, CRP, and SAA” has been added on Page 10 of the revised manuscript.
9. Sentences of “By the way, this proposed method also has the advantages of low cost and high precision compared with traditional dairy cow mastitis detection laws, such as microbiological diagnosis, somatic cell count and nuclear drug sensitivity test. Encouragingly,” and “National Key R&D Program of China (Grant No. 2021YFD2000804)” have been added on Page 12 of the revised manuscript.
10. Words of “1150f2” has been added on Page 13 of the revised manuscript.
11. Words of “Spectrochim. Acta. A Molec. Biomol. Spectrosc” has been rewritten on Page 14 of the revised manuscript.
12. The grammar has been improved and some other minor modifications have been made in a careful proof reading.

Reviewer 4 Report
This manuscript reports on development of SERS immunosensor for detection of mastitis in dairy cow. The importance of the problem related with development of highly sensitive and reliable sensors for detection of interleukin-6 (IL-6) biomarker is well-justified in the manuscript. The function of the sensor, the repeatability and sensitivity are clearly demonstrated. The important novelty of this manuscript is related with optimization of SERS nanotag response and assay conditions. The manuscript is very clear, well-written and conclusions are supported by experimentally observed findings. I recommend publication in Nanomaterials with minor revision considering the following points.
1) In my opinion, expression “dye” is not appropriate in the case of 4-mercaptobenzoic acid (4-MBA) because this compound does not exhibit absorption bands in the visible spectral region.
2) Please indicate in the revised version of manuscript important parameters of SERS experiment: laser power at the sample, objective magnification and numerical aperture (NA).
3) Reference [35]: please provide the page numbers.
4) Reference [45]: please check the abbreviation of the journal; should be “Spectrochim. Acta..”
Author Response
To Reviewer #4:
Comments: This manuscript reports on development of SERS immunosensor for detection of mastitis in dairy cow. The importance of the problem related with development of highly sensitive and reliable sensors for detection of interleukin-6 (IL-6) biomarker is well-justified in the manuscript. The function of the sensor, the repeatability and sensitivity are clearly demonstrated. The important novelty of this manuscript is related with optimization of SERS nanotag response and assay conditions. The manuscript is very clear, well-written and conclusions are supported by experimentally observed findings. I recommend publication in Nanomaterials with minor revision considering the following points.
Question 1: In my opinion, expression “dye” is not appropriate in the case of 4-mercaptobenzoic acid (4-MBA) because this compound does not exhibit absorption bands in the visible spectral region.
Answer: We thank for the reviewer’s valuable suggestion. Accordingly, we have replaced “dye” with “molecule” in Line 19, 89, and 214, and highlighted them in the revised manuscript.
Question 2: Please indicate in the revised version of manuscript important parameters of SERS experiment: laser power at the sample, objective magnification and numerical aperture (NA).
Answer: We thank for the reviewer’s valuable suggestion. Accordingly, we have added the sentence of “and the objective magnification is 20X with the numerical aperture of 0.5” and “with laser power of 10 mW” in Line 117 and 118, and highlighted them in the revised manuscript.
Question 3: Reference [35]: please provide the page numbers.
Answer: We thank for the reviewer’s good suggestion. We didn’t find the exact page number. Accordingly, we have added “1150f2” in Line 506 and highlighted it in the revised manuscript.
Question 4: Reference [45]: please check the abbreviation of the journal; should be “Spectrochim. Acta..”
Answer: We thank for the reviewer’s good question. We have rewritten “Spectrochim. Acta. A Molec. Biomol. Spectrosc.” in Line 528 and highlighted it in the revised manuscript.
List of Changes (Red in the context)
- All the modified graphics have been inserted in the revised manuscript.
- Words of “molecule” and “yuan” have been added on Page 1 in “Abstract” and “Introduction” section of the revised manuscript.
- Words of “glycoproteins” and “SPR”, and sentences of “These results are insufficient for effective diagnosis of inflammatory diseases in dairy cows. Therefore, it is urgent and challenging to establish a simple and easy method to quantitatively detect IL-6 in milk.” was added on Page 2.
- Words of “molecule” and sentence of “IL-8, IL-10, CRP (C-reactive protein), and SAA (Serum amyloid A) were purchased from Biodee Biotechnology Co., Ltd. (Shanghai, China).” , “and the objective magnification is 20X with the numerical aperture of 0.5” and “with laser power of 10 mW” has been added on Page 3 the revised manuscript.
- Words of “kV” has been rewritten on Page 4 of the revised manuscript.
- Words of “molecule” and sentence of “The IL-6 mAb antibodies were immobilized on the test region of nitrocellulose membrane by physical adsorption and used as the capture reagent as the capture reagent” and “After that, three times of washing was done to reduce the nonspecific adsorption of nanotags on the NC membrane.” have been added on Page 5 of the revised manuscript.
- Words of “antibody” has been added on Page 7 of the revised manuscript.
- Sentences of “IL-8, IL-10, CRP, and SAA” has been added on Page 10 of the revised manuscript.
- Sentences of “By the way, this proposed method also has the advantages of low cost and high precision compared with traditional dairy cow mastitis detection laws, such as microbiological diagnosis, somatic cell count and nuclear drug sensitivity test. Encouragingly,” and “National Key R&D Program of China (Grant No. 2021YFD2000804)” have been added on Page 12 of the revised manuscript.
- Words of “1150f2” has been added on Page 13 of the revised manuscript.
- Words of “Spectrochim. Acta. A Molec. Biomol. Spectrosc” has been rewritten on Page 14 of the revised manuscript.
- The grammar has been improved and some other minor modifications have been made in a careful proof reading.
